# Favorable Preclinical Pharmacological Profile of a Novel Antimalarial Pyrrolizidinylmethyl Derivative of 4-amino-7-chloroquinoline with Potent In Vitro and In Vivo Activities

**DOI:** 10.3390/biom13050836

**Published:** 2023-05-14

**Authors:** Nicoletta Basilico, Silvia Parapini, Sarah D’Alessandro, Paola Misiano, Sergio Romeo, Giulio Dondio, Vanessa Yardley, Livia Vivas, Shereen Nasser, Laurent Rénia, Bruce M. Russell, Rossarin Suwanarusk, François Nosten, Anna Sparatore, Donatella Taramelli

**Affiliations:** 1Dipartimento di Scienze Biomediche, Chirurgiche e Odontoiatriche (DiSBIOC), Università Degli Studi di Milano, Via Pascal 36, 20133 Milan, Italy; nicoletta.basilico@unimi.it; 2Dipartimento di Scienze Biomediche per la Salute, Università Degli Studi di Milano, Via Pascal 36, 20133 Milan, Italy; silvia.parapini@unimi.it; 3Dipartimento di Scienze Farmacologiche e Biomolecolari (DISFEB), Università Degli Studi di Milano, Via Pascal 36, 20133 Milan, Italy; sarah.dalessandro@unimi.it (S.D.); paola.misiano@guest.unimi.it (P.M.); 4Dipartimento di Scienze Farmaceutiche (DISFARM), Università Degli Studi di Milano, Via Mangiagalli 25, 20133 Milan, Italy; sergio.romeo@unimi.it; 5Aphad Srl, Via della Resistenza 65, Buccinasco, 20090 Milan, Italy; g.dondio@aphad.eu; 6Department of Immunology Biology, Faculty of Infectious and Tropical Diseases, London School of Hygiene and Tropical Medicine (LSHTM), Keppel Street, London WC1E 7HT, UK; vanessa.yardley@lshtm.ac.uk (V.Y.); livia.vivas.net@gmail.com (L.V.); shereen.nasser84@gmail.com (S.N.); 7Lee Kong Chian School of Medicine, Nanyang Technological University, Singapore 308232, Singapore; lcsrenia@ntu.edu.sg; 8A*STAR Infectious Diseases Labs, Agency for Science, Technology, and Research, Singapore 138648, Singapore; 9Department of Microbiology and Immunology, University of Otago, Dunedin 9054, New Zealand; b.russell@otago.ac.nz (B.M.R.); noi.suwanarusk@otago.ac.nz (R.S.); 10Shoklo Malaria Research Unit, Mahidol-Oxford Research Unit, Faculty of Tropical Medicine, Mahidol University, Mae Sot 63110, Thailand; francois@tropmedres.ac; 11Centre for Tropical Medicine and Global Health, Nuffield Department of Medicine, University of Oxford, Oxford OX3 7BN, UK

**Keywords:** 4-aminoquinolines, malaria, drug discovery

## Abstract

The 4-aminoquinoline drugs, such as chloroquine (CQ), amodiaquine or piperaquine, are still commonly used for malaria treatment, either alone (CQ) or in combination with artemisinin derivatives. We previously described the excellent in vitro activity of a novel pyrrolizidinylmethyl derivative of 4-amino-7-chloroquinoline, named **MG3,** against *P. falciparum* drug-resistant parasites. Here, we report the optimized and safer synthesis of **MG3**, now suitable for a scale-up, and its additional in vitro and in vivo characterization. **MG3** is active against a panel of *P. vivax* and *P. falciparum* field isolates, either alone or in combination with artemisinin derivatives. In vivo **MG3** is orally active in the *P. berghei, P. chabaudi,* and *P. yoelii* models of rodent malaria with efficacy comparable, or better, than that of CQ and of other quinolines under development. The in vivo and in vitro ADME-Tox studies indicate that **MG3** possesses a very good pre-clinical developability profile associated with an excellent oral bioavailability, and low toxicity in non-formal preclinical studies on rats, dogs, and non-human primates (NHP). In conclusion, the pharmacological profile of **MG3** is in line with those obtained with CQ or the other quinolines in use and seems to possess all the requirements for a developmental candidate.

## 1. Introduction

Drug resistance to artemisinin-based combination therapies (ACTs) is one of the factors hampering the success of the last twenty years of intervention to fight *P. falciparum (Pf)* malaria. From the year 2000, following the implementation of the Millennium Development Goals, and the economic and political commitment of local governments, international agencies and private funds, malaria mortality and incidence were reduced by 60% and 37%, respectively, with the total number of deaths declining from 896,000 in 2000 to 558,000 in 2019 [1]. In 2020 and 2021, a slight increase in malaria deaths was observed, in part because of the disruption of sanitary control due to the SARS-COV-2 pandemic. Most malaria cases and deaths occurred in the WHO African Region. However, from 2015–2020 the mortality reduction rate was slower than expected, indicating the need to reinforce both commitment and investments, but also to find new tools.

In terms of treatments, over the last ten years, partial resistance to artemisinin (due to PfK13 mutations of *P. falciparum* parasites) spread in SE Asia and emerged also in South America, and Africa [2,3,4,5,6,7]. In addition, resistance to the ACT partner drugs has also been observed, and both resulted in frequent treatment failures in the so-called Great Mekong Subregion [8,9]. If novel treatments to counteract drug-resistance are not introduced, the effort to eliminate and hopefully to eradicate malaria will be under significant threat. Moreover, elimination will be easier to achieve if, concomitantly, transmission blocking drugs, those active against the sexual gametocyte stages, will be identified [10]. 

Currently available drugs, with different targets and mechanism of resistance, have also been proposed as partner drugs with an artemisinin derivative in a triple combination therapy (T-ACT) [11]. Improved efficacy of T-ACT in vivo has been reported and may delay the emergence of resistance to any of its components [12]. The bottle neck is that very few effective antimalarials are available for T-ACT. Therefore, new products specifically designed for combination therapies need to be investigated.

The 4-aminoquinoline derivatives which include chloroquine (CQ) and amodiaquine, are characterized by a common scaffold, the 4-aminoquinoline, whose antimalarial properties are well documented [13]. Even if for most of these drugs resistant parasites have emerged, these molecules contributed to cure and save millions of lives over the years. In the last two decades, new molecules of this class, such as AQ13 [14,15,16] and ferroquine [17,18], reached clinical testing, indicating their validity. The latter, in combination, is currently in phase II for uncomplicated malaria [19]. The 4-aminoquinolines, however, cannot be considered proper transmission blocking antimalarials since their activity is usually limited to the killing of young, phase II–III stages of *Pf* gametocytes and not of mature stage V gametocytes, those able to infect *Anopheles* mosquitoes [20].

Moreover, every year between six and seven million cases of human malaria infections are due to *P. vivax* parasites, whose standard treatment is represented by CQ [21]. Resistance to CQ is present, but not at alarming levels, since treatment failure is observed in less than 5% of cases in most of *P. vivax* endemic countries [1]. Yet, activity against *P. vivax* parasites is also a desirable property of novel compounds to increase the limited armamentarium of available drugs.

For our research, we have synthesized and studied a series of quinolizidinyl derivatives of 7-chloro-4-aminoquinoline, one of which, named **AM-1** showed an excellent in vitro and in vivo antimalarial efficacy [22,23]. The presence of a bulky, strongly basic, and lipophilic bicyclic quinolizidine ring was an interesting structural feature of **AM-1,** able to overcome the resistance mechanisms by preventing the metabolic oxidative dealkylation, which affects the usual dialkylaminoalkyl chains of many CQ analogs. However, **AM-1** had a chiral structure and could be synthesized starting from the natural alkaloid *l-lupinine* or through a long and expensive enantioselective synthesis. Based on these promising results and in the attempt to find novel compounds easier and cheaper to be synthesized, we prepared and studied two novel pyrrolizidinylalkyl analogs, **MG2** and **MG3** [24], where a bulky, basic, and lipophilic head was still present, but linked in a position which prevents the chirality of the previous quinolizidinyl analogs [22,23].

Here, we extended the investigation on **MG3**, a pyrrolizidinylmethyl derivative of 4-amino-7-chloroquinoline, as a novel antimalarial candidate for further development (Figure 1). **MG3** belongs to the class of 5-substituted pyrrolizidine derivatives which includes biologically active substances used to treat chronic pathologies, such as the antiarrhythmic pilsicainide [25,26] or the nicotinic [27,28], muscarinic M1 and serotonin 5-HT4 receptor ligands useful in improving cognitive functions or as gastro-intestinal prokinetic agents [29,30,31,32]. 

Preliminary data indicated that **MG3** was very active in vitro against both CQ-Sensitive and CQ-Resistant strains of *P. falciparum* and in vivo in the mouse *P. berghei* model. Toxicity was low both in vitro and in vivo, with an excellent safety index [24]. Investigation on the mechanism of action of **MG3** was also performed using the in vitro assay BHIA (beta-hematin inhibitory activity) previously described [33]. **MG3** was able to inhibit the formation of beta-hematin (malaria pigment) and thus the detoxification of hemoglobin-derived haem, leading to the accumulation of toxic haem molecules in the parasite food vacuole, as already shown for CQ and several 4-amino-7-chloroquinoline derivatives. [24,33,34]. It also showed an interesting metabolic profile. The original synthesis, however, was not suitable for scaling up due to the use of hazard reactants and harsh reaction conditions. Moreover, the ADME/TOX profile of **MG3** needed further analysis.

Here, we report a novel scale-up synthesis of **MG3**, which is cheaper, safer, and more efficient compared to the previously described one. Non-formal, preclinical ADME/Tox profile of **MG3** and candidate selection has also been completed and additional activity data have been included. This novel 4-amino-7-chloroquinoline derivative may be further developed to phase I studies.

## 2. Materials and Methods

### 2.1. Chemistry

Chemicals and solvents were purchased from Merck, Sigma-Aldrich (Merck Sigma Italia, Milan, Italy) and used without further purifications. NMR spectra were recorded on a Varian 300 MHz instrument using CDCl_3_ or DMSO-*d*_6_ as solvent. TLC was carried out on Merck precoated 60 F254 plates using iodine for visualization. Organic phases were dried over anhydrous sodium sulphate. Concentrations were performed under reduced pressure (1–2 kPa) at a bath temperature of 40 °C. ESI-MS analyses were performed using a Thermo Finnigan (Waltham, MA, USA) LCQ Advantage system MS spectrometer with an electronspray ionisation source and an ‘Ion Trap’ mass analyzer. The MS spectra were obtained via direct infusion of a sample solution in methanol under ionization, ESI-positive. Analytical RP-HPLC was performed using Hitachi Elite LaChrom (Hitachi High-Tech Corporation, Tokyo, Japan) chromatography system, mobile phase A: phosphate buffer pH6, mobile phase B: methanol (for gradient details see Appendix A) and stationary phase: Purospher^®^ STAR RP-18 endcapped (5 µm) LiChroCART^®^ 250-4.

*1,7-dichloroheptan-4-one***(2).** Dicyclopropylketone (27 mL, 250 mmol) was cooled to 0 °C and HCl gas was bubbled. The reaction mixture was stirred at 25 °C for 4 h. At the end of the reaction (TLC monitoring) the excess of HCl was removed under vacuum and the crude product was purified via distillation (110 mTorr at 70 °C or 200 mTorr at 120 °C) to afford a colorless oil (43 g, 95%).

TLC: Rf = 0.65 (dichloromethane/cyclohexane 9:1). ^1^H NMR (300 MHz, CDCl_3_): δ 3.51 (t, 4H), 2.57 (t, *J* = 7.03 Hz, 4H), 1.98 (m, 4H). MS (ESI) *m/z* calcd for C_7_H_13_Cl_2_O [M+H]^+^: 183.03, found 183.06. 

*7a-(nitromethyl)hexahydro-1H-pyrrolizine***(4).** Compound **2** (43 g, 237 mmol) and nitromethane (45 mL, 950 mmol, 4.0 equiv) were dissolved in methanol (80 mL) and the reaction mixture was stirred at 25 °C for 72 h and bubbling ammonia was stirred for 4 h per day. At the end of the reaction (TLC monitoring), the ammonia evaporated under vacuum. NaOH (200 mL, 0.1 M) was added to the crude product and the aqueous phase was extracted with dichloromethane (6 × 50 mL). The organic layer was dried with Na_2_SO_4_, filtered, and concentrated under vacuum to give a colorless oil (38 g, 95%), pure enough to be use in the following reaction.

TLC: R_f_ = 0.58 (dichloromethane/methanol 9:1). ^1^H NMR (300 MHz, CDCl_3_): δ 4.27 (s, 2H), 3.02–2.91 (m, 2H), 2.60–2.47 (m, 2H), 2.12–1.92 (m, 2H), 1.89–1.49 (m, 6H). MS (ESI) *m/z* calcd for C_8_H_15_N_2_O [M+H]^+^: 171.11, found 171.14.

*1-(tetrahydro-1H-pyrrolizin-7a(5H)-yl)methanamine***(5).** To a solution of NaBH_4_ (69 g, 1.8 mol, 7.2 equiv) in ethanolic NaOH (2%, 520 mL), a suspension of palladium on carbon (10%, 12 g) in water (175 mL) was slowly added under nitrogen. Then, a solution of **4** (38 g, 225 mmol) in methanol (500 mL) was added to the suspension dropwise. The reaction mixture was stirred at 25 °C for 12 hours. At the end of the reaction (TLC monitoring), the mixture was filtered, and the solution obtained was concentrated under vacuum. Saturated NaHCO_3_ solution (200 mL) was added to the crude product and the aqueous phase was extracted with diethyl ether (6 × 50 mL). The organic layer was dried with Na_2_SO_4_, filtered, and concentrated under vacuum. The crude product was purified via distillation (7.5 Torr at 150 °C) to afford a colorless oil (13.2 g, 42%).

TLC: R_f_ = 0.20 (dichloromethane/methanol/NH_4_OH 8:2:0.2). ^1^H NMR (300 MHz, CDCl_3_): δ 2.93–2.84 (m, 2H), 2.57–2.47 (m, 2H), 2.44 (s, 2H), 1.79–1.55 (m, 8H), and 1.50–1.38 (m, 2H). MS (ESI) *m/z* calcd for C_8_H_17_N_2_O [M+H]^+^: 141.14, found 141.15.

*7-Chloro-N-((hexahydro-1H-pyrrolizin-7a-yl)methyl)quinolin-4-amine* 
**(MG3**
*), method A*

A mixture of (hexahydro-1*H*-pyrrolizin-7a-yl)methanamine (1.5 g, 10 mmol), 4,7-dichloroquinoline (1.98 g, 10 mmol) and phenol (0.941 g, 10 mmol) was heated for 4 h at 120 °C, stirring under nitrogen. After cooling to room temperature, the mixture was alkalized with 2 N NaOH and extracted thoroughly via CH_2_Cl_2_. The organic extracts were sequentially washed with 2 N NaOH, water and finally with 5% acetic acid.

The acetic solution was alkalized with 2N NaOH and the suspension was warmed to 40 °C to favor the flocculation. After filtration, the product was washed with water and then with ether and kept overnight in a desiccator. **MG3** (1,97 g, yield: 65%) as white solid with m.p. 110–111.5 °C ([24] lit. 112–113 °C) was obtained. Purity > 99% (HPLC) (Appendix A).

^1^H NMR (300 MHz, DMSO-*d_6_*): δ 8.37 (d, *J* = 5.3 Hz, 1H), 8.22 (d, *J* = 9.1 Hz, 1H), 7.78 (bd, 1H), 7.43 (d, *J* = 9.1 Hz, 1H), 6.86 (bt, 1H), 6.62 (d, *J* = 5.4 Hz, 1H), 3.15 (d, *J* = 5.3 Hz, 2H), 2.97 2.90 (m, 2H), 2.58-2.49 (m, 2H), 1.92-1.61 (m, 6H), and 1.57-1.49 (m, 2H). (Appendix A). COSY NMR consistent with structure (Appendix A). ^13^C NMR (75 MHz, DMSO-*d_6_*): δ 152.30, 151.14, 149.55, 133.80, 127.97, 124.52, 124.18, 117.83, 99.57, 74.06, 55.66, 51.10, 36.40, and 25.60. (Appendix A).


*7-Chloro-N-((hexahydro-1H-pyrrolizin-7a-yl)methyl)quinolin-4-amine (*
**MG3**
*), method B*


A mixture of (hexahydro-1*H*-pyrrolizin-7a-yl)methanamine (3 g, 20 mmol), 4,7-dichloroquinoline (1.98 g, 10 mmol) was heated for 5 h at 120 °C, stirring under nitrogen. After cooling, the mixture was stirred with water (13 mL) and 2N NaOH (5 mL) and filtered. The solid residue was again thoroughly washed with water and the aqueous washes were pooled, saturated with Na_2_CO_3_, and extracted several times with CH_2_Cl_2_ to recover the unreacted amine (1.4 g). The solid residue was dissolved in CH_2_Cl_2_ and extracted with 5% acetic acid. The acetic solution was alkalinized with 2 N NaOH and the suspension was filtered first with water and then with ether to obtain **MG3** (1.7g, m.p. 109–111 °C). From the ether washings gathered and evaporated to dryness a solid residue is obtained which, after washing with water and then with ether, resulted in 234 mg of **MG3**. From the mother liquor evaporated to dryness and washed again, additional 220 mg of **MG3** have been obtained. Yield: 71.4%. Purity > 98% (HPLC) (Appendix A).

^1^H NMR (300 MHz, DMSO-*d_6_*): δ 8.37 (d, *J* = 5.4 Hz, 1H), 8.22 (d, *J* = 9.1 Hz, 1H), 7.79 (s, 1H), 7.43 (d, *J* = 9.0 Hz, 1H), 6.87 (bt, *J* = 6.8 Hz, 1H), 6.63 (d, *J* = 5.4 Hz, 1H), 3.16 (d, *J* = 5.6 Hz, 2H), 2.98–2.91 (m, 2H), 2.58–2.50 (m, 2H), 1.90–1.64 (m, 6H), and 1.62–1.50 (m, 2H). (Appendix A) COSY NMR consistent with structure (Appendix A). ^13^C NMR (75 MHz, DMSO-*d_6_*): δ 152.32, 151.16, 149.54, 133.83, 127.95, 124.56, 124.20, 117.83, 99.58, 74.18, 55.66, 51.06, 36.39, and 25.59 (Appendix A).


*7-Chloro-N-((hexahydro-1H-pyrrolizin-7a-yl)methyl)quinolin-4-amine di-mesylated salt*


To an iced ethanolic solution of **MG3** (100 mg, 0.33 mmol) methansolfonic acid (0.04 mL, 0.66 mmol) was added and the solution was evaporated to dryness. The residue was again dissolved in absolute ethanol and ether was added dropwise until complete precipitation of the salt, which was filtered and dried. An off-white powder with m.p. 210–211.8 °C was obtained.


*7-Chloro-N-((hexahydro-1H-pyrrolizin-7a-yl)methyl)quinolin-4-amine dihydrochloric salt*


To an iced ethanolic solution of **MG3** (558 mg, 1.84 mmol) an 1N ethanolic HCl (4 mL, 4 mmol) was added, and the solution was evaporated to dryness. The residue was again dissolved in absolute ethanol and evaporated to dryness and this procedure was repeated several times to eliminate water. Finally, the solid residue was rinsed with ether, filtered, and dried at 40 °C under vacuum. M.p. 276–280 °C (dec).

### 2.2. Biological Assays

#### 2.2.1. *P. falciparum* Cultures and Drug Sensitivity Assays

The strains D10 and 3D7 (CQ-Sensitive), and V1-S (CQ, pyrimethamine and cycloguanil resistant), K1 (CQ and PYR resistant), W2 and FCR3 (CQ-Resistant), TM90C2A (CQ, MQ, PYR and ATQ resistant) [35] and IPC 5202 (MR4-1240, artemisinin-resistant, K13 mutation, and R539T) [7] of *Pf* were cultured in vitro as described by Trager and Jensen with minor modifications [36]. Briefly, parasites were maintained at 5% hematocrit (human type A-positive red blood cells) in RPMI 1640 (EuroClone, S.p.A. Milan, Italy) medium with the addition of 10% heat inactivate human serum or 1% AlbuMax (In vitrogen, Milan, Italy), 20 mM Hepes and 2 mM glutamine (Euroclone), at 37 °C in a standard gas mixture consisting of 1–3% O_2_, 5% CO_2_, and 92–94% N_2_.

Compounds were dissolved in either water or DMSO and then diluted with medium to achieve the required concentrations (final non-toxic DMSO concentration <1%). Asynchronous cultures of *Pf* with parasitaemia of 1–1.5% and 1% final hematocrit were aliquoted into 96-well flat-bottom microplates (Corning, Glendale, AZ, USA) with serial dilutions of test compounds and incubated for 72 h at 37 °C. Parasite growth was determined spectrophotometrically (OD_650_) by measuring the activity of the parasite lactate dehydrogenase (pLDH) according to a modified version of the method of Makler in control and drug-treated cultures [37]. The antimalarial activity is expressed as 50% inhibitory concentrations (IC_50_); each IC_50_ value is the mean and standard deviation of at least three separate experiments performed in duplicate [22].

#### 2.2.2. Field Isolates of *P. falciparum* and *P. vivax* and Chemosensitivity Assays

The 10 *P. vivax* and 10 *P. falciparum* isolates used in this study were obtained from Mae Sod District, Tak Province, Thailand. Samples were collected from patients attending the clinics of the Shoklo Malaria Research Unit (SMRU) from June to July 2010 under the ethical guidelines (IRB OxTREC 58-09 and 04-10). Isolates were collected in 5 mL lithium-heparinized tubes and sent to the laboratory within 5–6 h. After platelets and leukocytes removal, the susceptibility of these isolates to **MG3**, chloroquine diphosphate (CQ) (Sigma-Aldrich) and artesunate (ASN) (Holly Pharmaceuticals Co Ltd., Beijing, China) was assessed as previously described [38].

#### 2.2.3. In Vivo Oral Efficacy Studies

In vivo antimalarial efficacy tests were performed at LSHTM using *P. berghei* ANKA (CQ-S)*, P. chabaudi chabaudi* AS and *P. yoelii yoelii NS* (CQ-R) rodent species in the standard four-day suppressive Peters’ test [39]. Groups of five CD1 mice were inoculated i.v., with 4 × 10^6^ infected erythrocytes/mice. Compounds, dissolved in standard suspending formula (SSV) (0.5% sodium carboxymethylcellulose, 0.5% benzyl alcohol, 0.4% Tween 80, and 0.9% NaCl), were administered once a day for 4 days via oral gavage, starting two hours post-infection. In order to determine ED_50_ and ED_90_ values, four drug doses were used (from 1 to 30 mg/kg). Parasitemia was evaluated using microscopic examination of Giemsa-stained blood films taken on day four, processed using MICROSOFT^®^ EXCEL spreadsheets (Microsoft Corp., Redmond, WA, USA), and expressed as percentages of inhibition from the arithmetic mean parasitemia of each group in relation to the untreated group. ED_50_ and ED_90_ values were calculated using GraphPad Prism 6 (GraphPad Software Inc., La Jolla, CA, USA). All animal work was carried out under a UK Home Office project license according to the Animal Act 1986 and the European Directive 2010/63/EU. The project license (PPL70/8427 was reviewed by the LSHTM Animal Welfare and Ethical Review Board prior to submission and consequent approval by the UK Home Office.

### 2.3. In Vitro Pre-Clinical Studies

#### 2.3.1. Intrinsic Clearance

For the intrinsic clearance determination, MG3.2HCl (1 µM) was incubated with mouse, rat, dog, monkey, and human hepatocytes (Celsis In Vitro Technologies Inc., Baltimore, MD, USA) in a final incubation volume of 0.75 mL Leibovitz L-15 medium, at 37 °C, in duplicate in 48-well plates under shaking. An automatic liquid handling system (Multiprobe II EX, Packard) was used for sampling. At 0, 10, 20, 30, 60 and 90 min, 50 mL aliquots of the incubates were taken and added with 80 mL of ice-cold ACN and 20 mL of 1 µM warfarin in ACN (internal standard), and samples centrifuged at 2000 rpm for 20 min. The analysis of supernatant was conducted using LC-MS/MS. **MG3** was incubated in the medium under the same conditions to evaluate the chemical stability; 7-ethoxycoumarin (7-ETC) 1 µM and 7-hydroxycoumarin (7-HC) 30 µM were utilized as positive controls for Phase I and Phase II activities. The intrinsic clearance (CLint) of **MG3** and of 7-ETC was calculated using the half-life approach. Half-life and CLint were determined from the concentration (counts) remaining at the different sampling points using the LC-MS/MS method. The slope was calculated using linear regression analysis of the curve (natural logarithmic area of the compound remaining against the time) and converted into the half-life (t1/2) and CLint expressed as µL/min/million cells and mL/min/kg, respectively.

#### 2.3.2. Metabolic Profiling and Metabolites Identification

For metabolite profiling, MG3.2HCl (10 µM) was incubated with mouse, rat, dog, monkey, and human hepatocytes (300,000 cells/mL for the mouse and 1 × 10^6^ cells/mL for the other species) in Leibovitz L-15 medium, at 37 °C. Incubations were performed in a 12-well plate under shaking. Aliquots of the incubation samples, together with a control in medium alone, were taken at t = 0 and 60 min; the metabolism was stopped by the addition of an equal volume of chilled ACN and the samples were centrifuged at 1100 rpm for 20 min. The supernatant was stored at −20 °C until analysis. The analyses were performed using an HPLC system (Agilent) online with a photodiode array (PDA) detector (Agilent) coupled to a Mass Spectrometer Q-TOF 2 (Waters), ESI positive, following the method reported in Table 1.

The software utilized for searching the accurate masses of possible metabolites was Metabolynx, version 4.1 (Waters Corporation, Milford, MA, USA), with a mass window of 40 mDa. Samples were then re-run in MS/MS mode by selecting the masses of possible metabolites found. The collision energy used to obtain MS/MS data was 25 eV. The proposed identity of metabolites was based on their accurate mass (using LockSpray correction), with an accuracy of +/−5 mDa. MS/MS analyses were performed to verify that peaks were drug-related and to obtain structural information from the fragment ions formed. The structures of some metabolites are proposed. The approximate relative amounts of parent and metabolites were determined from the absolute areas taken from a selected ion chromatogram at the metabolite masses with a mass window of 0.04 Da. Due to possible differences in response factors for each metabolite, these results should be regarded as semi-quantitative (approximate relative amounts of **MG3** and each metabolite in the samples).

#### 2.3.3. P450 Assay

The BD Gentest assays (Becton Dickinson Italia S.p.A Milan, Italy) with specific substrates that become fluorescent upon CYP metabolism were utilized to evaluate the potential inhibition of the five most important human P450 isoforms (CYP1A2, CYP2C9, CYP2C19, CYP2D6, and CYP3A4). **MG-3** and CQ were dissolved in DMSO and tested at 3 µM in duplicate (n = 2–3) in a 96-well plate [40]. Controls, known as selective isoform inhibitors, were added in each experiment. Data were expressed as percentage inhibition in respect to the control without inhibition (<5% inhibition was considered no effect), and for standards, the IC_50_ (concentration causing 50% inhibition) was calculated using Grafit v. 5.0.1 (Erithacus Software Ltd., Surrey, UK).

### 2.4. In Vivo Preclinical Studies

All the in vivo experiments described in the following sections were conducted using the CRO Accelera (Nerviano, Italy)). As stated by the CRO, all the environmental conditions, as well as all the procedures adopted through the study for housing and handling the animals are in strict compliance with EC and Italian Guidelines for Laboratory Animals Welfare. Statistical analysis was not performed due to the low number of samples per group.

#### 2.4.1. Pharmacokinetic Studies in Rat

The formulations were prepared on the days of dosing, immediately before administration. **MG3** for IV administration was formulated as in situ dimesylate salt, in 5% of glucose solution (4 mg/mL nominal concentration), while for oral administration it was prepared by suspending the test item in 0.5% methocel solution (*v*/*v*) (6 mg/mL nominal concentration). Male Sprague Dawley rats (Charles River, Italy) underwent SVC (superior vena cava) cannulation under anesthesia (by a mixture of ketamine, xylazine and acepromazine). After 5-days postoperative recovery period, **MG3** was administered and serial blood samples (about 0.3 mL) were collected from femoral vein using heparinized syringe at 5, 15, and 30 min, 1, 3, 6, 24-, 48-, 72- and 96-hours post-dosing after IV administration and at 15, and 30 min, 1, 3, 6 and 24-, 48-, 72- and 96-hours post-dosing after oral administration. The samples were then transferred into pre-cooled tubes protected from light and centrifuged at 3000 rpm for 4 min at +4 °C to collect plasma, and single aliquots of plasma were stored at −80 °C until analysis. Sample analysis: 400 µL of methanol was added to 10 µL of plasma in a 96-well plate; after centrifugation for 15 min at 2000 rpm at 6 °C, an aliquot of 360 µL of supernatant was transferred into a new 96-well plate and dried at 40 °C. Sample extracts were reconstituted with a volume of 150 µL of a mixture solution of 10 mM ammonium formate pH 3.5:methanol (90:10; *v*/*v*) and centrifuged for 3 min at 2000 rpm at 6 °C and the supernatant was injected onto the LC-MS/MS system. A HP1100 HPLC with a Synergy 4 um Hydro-RP 80A 2.0 × 50 mm column, coupled with AB/MDS Sciex 4000 QTrap in positive ion mode was utilized for analysis; 10 µL samples were injected, a gradient with 10 mM ammonium formate pH 3.5 and methanol was utilized. Data obtained were analyzed using the non-compartmental approach (Watson package v. 6.4.0.04, Thermo Fisher Scientific, Waltham, MA, USA).

#### 2.4.2. Tissue Distribution in Rat

Female Sprague Dawley rats (Charles River, Italy) (2–4/time points) were orally administered with a single dose of 120 mg/kg of **MG3** at a constant dose volume of 10 mL/kg; dose and gender were selected based on previous toxicological studies. Two female rats were included as untreated controls. Plasma samples and tissues, i.e., adrenal glands, hearth, kidney, liver, lung, lymph nodes, spleen, ovaries, thymus, and skeletal muscle were collected at 4, 24, 48, 72 and 96 h after administration by animals killed by exsanguination under sodium thiopental anesthesia. Blood was put in heparinized plastic tubes kept on ice-water bath, then centrifuged (10 min, 1200 rpm, +4 °C). One aliquot of about 100 µL of plasma was stored at –80 °C until analysis. Samples of selected organs/tissues were collected, put in plastic cassettes and frozen over liquid nitrogen. Then, organs were thawed out, weighted, and sliced. Homogenates were prepared with ultrasonication on ice in Dulbecco buffer (10 μL/mg of tissue). The addition of 400 μL of methanol to 10 μL of homogenate in a 96-well plate allowed the precipitation of proteins. After capping and vortex mixing, the plate was centrifuged for 15 min at 3700 rpm at 6 °C, and an aliquot of 360 μL of supernatant transferred to a new plate and dried down under nitrogen stream (40 °C). The extract was reconstituted in 150 μL of 10 mM ammonium formate pH 3.5: methanol (90:10, *v*/*v*) and analyzed using the LC-MS-MS method, as described in Pharmacokinetic studies in rat paragraph. 1 ng/mL and 10 ng/g were the lower limits of quantitation for plasma and tissue, respectively.

#### 2.4.3. Plasma Protein Binding and Blood to Plasma Partitioning

Plasma protein binding was performed via ultrafiltration (30,000 D cutoff, Amicon, Merck) with heparinized blood samples (male SD rats, Charles River Italy, and male Beagle dogs, Marshall Farms, Europe, Lyon, France) and fresh and frozen heparinized human blood samples collected from fasted healthy male subjects (at least two subjects for plasma samples), were supplied by AAI, Germany. **MG3** final concentrations of 100, 300 and 1000 ng/mL were prepared using a 10 mg/mL stock solution while control samples at the same concentrations diluted in DPBS buffer (Dulbecco’s Phosphate Buffered Saline, Sigma-Aldrich) to evaluate the recovery. After vortex-mixing, triplicate aliquots of plasma and buffer samples (about 1 mL) were loaded into the sample reservoir of the Centrifree system, and then centrifuged at 1500× *g* for 10 min at 37 °C. The percentage of bound drug (% F_bound_) and of unbound drug (% F_free_) was determined using the following equations: % F_bound_ = [(Conc. Total–Conc. Free)/Conc. Total] * 100 and % F_free_ = 100 − % F_bound_, where Conc. Total is the compound concentration determined in plasma and Conc. Free is the compound concentration determined in the ultrafiltrated samples. Results were expressed as % of binding and % of free fraction (single values and mean ± SD). Determination of compound concentrations was performed using the LC-MS/MS method. In blood to plasma partitioning experiments, the working solutions were added to rat, dog and human blood to obtain a final concentration of 300 ng/mL, immediately mixed and incubated at 37 °C under stirring; aliquots were collected at T0, 5, 10 and 30 min: an amount of blood samples was prepared for analysis using liquid–liquid extraction and an amount was centrifuged (3000 rpm for 3 min at +4 °C) to harvest plasma; whole blood and plasma samples were stored at –80 °C until analysis using the LC-MS/MS for test compound concentration. Blood to plasma ratio was calculated as: Conc. in blood/Conc. in plasma; where, Conc. is the compound concentration determined in blood and in plasma at the same time of incubation.

### 2.5. Toxicology

#### Dose Range Finding (DRF)

Seven days oral DRF in rats and dog, and five days oral DRF in Non-Human Primates (NHP) were performed using Sprague Dawley rats (n = 3/time point) of both sexes, one male and one female dog Beagle (Marshall/Green Hill) and one male and one female monkeys/Cynomolgus (Noveprim Ltd., Mahebourg Mauritius,). **MG3** was administered as oral suspension in 0.5% methocel in water (400 cP viscosity) at the following doses: 30, 60 and 120 mg/kg/day for 7 days (rat), 15 and 30 mg/kg/days for 7 days (dog) and 25, 50 and 75 mg/kg/day (NHP). No Observed Effects Levels (NOEL) were determined. Mortality, behavior, and general conditions were observed daily. Body weight was recorded pre-test, and on Days 1, 4 and 7. Food consumption was recorded from Days 1 to 4 and 4 to 7. Hematological and clinical chemistry examinations were performed on Day 8. All animals were sacrificed under anesthesia at the end of the treatment period on Day 8 of study. Post-mortem examinations included necropsy, collection, and histological examination performed of selected organs/tissues. 

Systemic exposure was measured in three additional rats/sex/group at different time-points up to 24 and 48 h after the treatment and on Days 1 and 7, respectively.

## 3. Results and Discussion

### 3.1. Chemistry: Novel Synthesis and Chemical Characterization of MG3

**MG3** is a pyrrolizidinylmethyl derivative of 4-amino-7-chloroquinoline, not chiral, synthesized in low gram scale from the commercially available intermediates (hexahydro-1H-pyrrolizin-7a-yl)methanamine and 4,7-dichloroquinoline in presence of phenol, as previously reported [24]. However, since (hexahydro-1H-pyrrolizin-7a-yl)methanamine is very expensive, **MG3** was obtained by a reduction in the corresponding nitrile, which in turn was formed by cyclization of 1,7-dichloro-4-heptanone with NH_3_ and 2-amino-2-methylpropanenitrile (Figure 1).

To address the need for large quantities of **MG3** for in vivo studies, the original synthetic route has been revised to avoid costly and/or dangerous steps, harsh reaction conditions, and chromatographic purifications. Indeed, the 2-amino-2-methyl-propanenitrile is not easily available from commercial sources and it is expensive; on the other hand, the synthesis of 2-amino-2-methyl-propanenitrile utilizes a toxic reagent, such as sodium or potassium cyanide [41,42,43,44], or acetone cyanohydrin [25], which are not suitable for a large laboratory or an industrial scale-up. The synthetic approach we used for a large laboratory scale-up synthesis (from 20–30 mmol to 250 mmol scale) of **MG3** is described in Figure 2. A further scale-up should not be difficult from the new available procedure. 

1,7-Dichloroheptan-4-one **2** was obtained in excellent yields (95%) from commercially available dicyclopropylketone via bubbling HCl followed by distillation. The 5-substituted 1-azabicyclo [3.3.0]octane **4** was prepared according to Oka et al. [27,29], thus the iminium intermediate **3** was generated in situ treating dichloroheptanone **2** with gaseous ammonia, followed by the nucleophilic addition of nitromethane to give the desired product. To reduce the formation of the by-product (dihydropyrrole), it was necessary to stir the mixture for 72 h and bubble ammonia for 4 h per day. Through this procedure, we obtained a large amount of compound **4** in excellent yields (95%) and sufficiently pure to be used in the following step without chromatographic purification. The reduction in the nitro group to amine **5** was the critical point of the scale-up synthesis considering the hazard associated with the use of Ni-Raney or lithium aluminum hydride. Therefore, we developed a mild procedure that allowed the reduction in the nitro group with sodium borohydride in the presence of 10% of palladium on carbon wet. Amine **5** was thus obtained on a 250 mmol scale, after distillation, in acceptable yields (42%). In the last step, the (hexahydro-1*H*-pyrrolizin-7a-yl)methanamine **5** reacted with 4,7-dichloroquinoline at 120 °C (instead of 180 °C) in the presence of only one equivalent of phenol (instead of the 7 eq previously used) to give **MG3** (65% yield). Note that the use of phenol can be avoided if two equivalents of the amine **5** are used (Figure 2, e) method B). Attempts to perform the reaction in organic solvents, such as isopropanol, n-butanol, toluene, and N-methyl pyrrolidone, in presence of K_2_CO_3_ (to trap the acid developed) afforded much lower yields (Appendix A).

### 3.2. In Vitro and In Vivo Characterization of MG3

#### 3.2.1. In Vitro Antimalarial Activity against *P. falciparum* Asexual Stages

The in vitro activity of **MG3** against different *P. falciparum* strains was independently determined at the University of Milan and at LSHTM (UK), as described in the M&M section. **MG3** confirmed its strong activity against both CQ-sensitive (CQ-S) and CQ-resistant (CQ-R) parasites with an IC_50_ in the low nanomolar range with a mean IC_50_ of 20.9 nM (range 12–30 nM) against all the *Pf* strains tested. No signs of cross reactivity with CQ were seen. As expected for this class of compounds, the activity of **MG3** was 5–6 fold lower than that of ASN or DHA. However, **MG3** maintained very high activity (IC_50_ < 15 nM) against the k13 mutant *P. falciparum* IPC 5202, artemisinin resistant strain, and 10 different *P. falciparum* field isolates. Moreover, against *P. vivax* isolates, **MG3** showed an IC_50_ 40% lower than that of CQ (50.3 nM vs. 86.6 nM), but not as good as that of artesunate (2.3 nM) (Table 2, right two columns). 

Compared to the quinolizidine-modified 4-aminoquinoline, AM-1, that we described earlier [23], **MG3** shows similar activity in vitro against drug-resistant *Pf* strains, but has the advantage of being achiral, and is easier to synthesize. Overall, the in vitro activity of **MG3** against asexual stage of *Pf* strains is comparable or superior to that of other novel 4-aminoquinolines including isoquine, N-tert-butyl isoquine or FAQ-4 [45] or piperaquine (IC_50_ = 21.2 vs. 49.0 nM against K1 strain) [46]. In vitro combination studies of **MG3** with ASN, DHA, or RK182 [47] revealed additive interaction, with a non-significant trend toward antagonism (see Appendix A [48,49]). Taken together with the potent activity against *Pf* IPC5202, ARM-R strain, these results suggest that **MG3** could be a suitable partner drug for combination treatments. Moreover, and as expected for this class of molecules, **MG3** was not a potent transmission blocking compound since its activity against young, stage I-III gametocytes was around 100 nM, such as CQ, but it decreased against mature stage V gametocytes (Appendix A). Methylene Blue, utilized as positive control, maintained its potency in both stages [20,50,51].

#### 3.2.2. In Vivo Oral Efficacy of MG3

The in vivo efficacy of **MG3** was evaluated in three different murine infection models using the standard Peter’s 4-day test. Efficacy studies were conducted in mice infected with *P. berghei* ANKA, *P. chabaudi AS* or the CQ-R strain *P. yoelii NS.* The results indicated a good oral activity of **MG3** which was slightly superior to that of FAQ-4, a 4′-Fluoro-N-tert-butylamodiaquine (2k) [45] or CQ tested in parallel (Table 3). The ED_50s_ and ED_90_ obtained against the three different species confirmed the preliminary data previously reported against *P. berghei* [24].

**MG3** was not curative when given at 30 mg/kg for 3 times to *P. berghei*-infected mice, but it prolonged the mice survival by approximately 50% compared to CQ, amodiaquine or artesunate (Appendix A).

### 3.3. Characterization of ADME-TOX Profile of MG3

#### 3.3.1. Intrinsic Clearance, Metabolism and P450 Interactions of MG3

The half-lives of **MG3** in mouse, rat, dog, monkey, and human hepatocytes and the corresponding intrinsic clearance values are reported in Table 4. The clearance of **MG3** increased from man > rat and monkey > mouse and dog. These results are comparable to those reported for CQ [52], but different from isoquine [53]. The slope of the concentration of compound remaining against time was linear up to 90 min in all species. **MG3** incubated for 90 min at 37 °C with the incubation medium alone did not show any degradation.

The relative amounts of **MG3** and metabolites detected after 1 h incubation with hepatocytes of different species are reported in Table 5. Unchanged **MG3** was the major component in man and rat, accounting for 82% and 53% of total drug-related material, respectively, whilst in mouse, dog, and monkey **MG3** accounted for about 30%. In all species, except dogs, the major metabolite was **M1**, accounting for 44, 24, 17 and 11% in monkey, mouse, rat, and human, respectively (Table 5 and Figure 2). In dogs, **M1** accounted for 10%. According to the elemental composition and MS/MS spectrum, **M1** (*m*/*z* = 316.12), had a keto group in the pyrrolizidinyl moiety, but the exact position could not be assigned. The main metabolite in dogs (27%) was **M2**. In mouse, rat, monkey, and human, **M2** accounted for 3, 6, 1 and 1%, respectively. **M2** (*m*/*z* = 318.14), was hydroxylated on the alkyl bridge.

Four additional mono-hydroxy metabolites with *m*/*z* = 318.14 were detected. **M3** accounted for 11% in dog, while in the other species was in the range <1–4%. **M6** accounted for 12–15% in mouse and rat, while in the other species was in the range of 13%. **M4** (mouse only) and **M5** (mouse and rat) were detected in traces (about 1%). From the fragmentation spectra, **M3** was hydroxylated in the pyrrolizidinyl moiety, while **M4, M5** and **M6** in the chloroquinoline moiety. Additional hydroxylation of **M3** afforded metabolite **M8** (*m*/*z* = 334.13), accounting for 2, 4 and 9% in mouse, dog, and monkey, respectively; **M8** was not detected in rat and human. With mouse, rat, dog, and monkey hepatocytes three di-oxidated metabolites with *m*/*z* = 332.12 were detected (**M9**, **M10** and **M11**), in amounts <1–4%. The fragmentation spectra and the extensive loss of water indicated that the hydroxy group in **M9** and **M10** was on the pyrrolizidinyl moiety. **M11** did not show loss of water, thus indicating that the hydroxy group was in the chloroquinoline moiety. In animal species, low amounts (<2%) of a tri-oxidated metabolite (**M12**, *m*/*z* = 348.11) were also detected. Metabolites **M13** and **M14** showed *m*/*z* = 358.17 and metabolite **M17** showed *m*/*z* = 388.15. From the fragmentation spectra, the metabolic biotransformation occurred in the pyrrolizidinyl moiety, but no structure could be assigned; only the elemental composition was given. **M14** ranged from <1% (human) to 9% (dog); **M13** and **M17**, when present, were in low amounts (<2%). Metabolites **M15** and **M16** (*m*/*z* = 390.16) showed loss of water in the MS/MS spectra, thus suggesting the presence of a hydroxy group in the pyrrolizidinyl moiety, but no structure could be assigned. Both metabolites were detected in low amounts (<2%) in the animal species, but not in human. Metabolite **M18** (*m*/*z* = 434.18) was a conjugate (+132 Da) of unchanged **MG3**. The exact position of conjugation could not be assigned from the fragmentation spectrum. Around 2% of this metabolite was detected in monkey and human and also in rat and dog. The MS/MS spectra of **MG3**, of metabolites **M1, M2, M3, M6** and **M8** are shown in Appendix A. In conclusion, the metabolism of **MG3** occurred mainly through phase I reactions. Inter-species differences in the metabolite profiles were observed only from a quantitative standpoint. In all species, except dog, the major metabolite had a keto group in the pyrrolizidinyl moiety; this metabolite ranged from about 10% (dog and human) to 44% (monkey). In dog, the major metabolite (27%) was hydroxylated on the alkyl bridge; in mouse, rat, monkey, and human this metabolite was detected in amounts lower than 6%. Mono and di-hydroxylation of **MG3** in the pyrrolizidinyl moiety were main metabolic pathways in dog and monkey; in mouse, rat and human, hydroxylation in the chloroquinoline moiety of **MG3** was preferred. All the metabolites identified after incubation with human hepatocytes were also detected with hepatocytes of the animal species.

The results from the **MG3** metabolism study confirm, as expected, that no potential toxic pyrrole metabolites are formed. This was a matter of concern since it is well known that the hepatotoxicity of the pyrrolizidine alkaloids found in many *Senecio*, *Heliotropium* and *Crotolaria* species as well as the toxicity among pyrrolizidine alkaloids varies dramatically with differences in chemical structure. Several structural requirements for toxicity have been identified and summarized by McLean [54]. It is confirmed that the toxicity is confined to the unsaturated alkaloids, which can be transformed into highly reactive pyrrole intermediates upon metabolism by CYP3A4. A double bond between carbon 1 and 2 in the pyrrolizidine ring system is essential for the toxic effects of the alkaloid, along with additional esterified hydroxyl groups with at least one containing a branched carbon chain around the nucleus [55]. However, the substitution at the bridgehead carbon (position 5), that characterizes compound **MG3**, does not allow the formation of pyrrole-containing metabolites, in accordance with our findings. 

The interaction of **MG3** with the major human P450 isoenzymes was investigated using the Fluorescent High Throughput P450 assay. As shown in Table 6, the interaction of **MG3** at the fixed concentration of 3 µM with the P450 CYP isoforms was very low, especially with the isoforms 2D6 and 3A4 that together are responsible for the metabolism of about 60% of the most clinically important drugs. In the same conditions, CQ showed a significantly higher inhibition of CYP2D6, as recently confirmed [56]. Assuming a possible use of **MG3** in combination therapy, from the data in the literature [57] it appears that artesunate and DHA partially inhibit CYP1A2, 2B6, 2C19 and 3A4. A high risk of interaction in vivo was predicted if artemisinin is co-administrated with CYP1A2 or 2C19 substrates. Therefore, **MG3** should not interfere extensively with the metabolism of other drugs potentially used in combination chemotherapy.

#### 3.3.2. In Vivo Pharmacokinetics Study in Rats

Pharmacokinetic study of **MG3** and its absolute bioavailability were performed after IV (10 mg/kg) or oral (30 mg/kg) compound administration in male albino Sprague Dawley rats. Following IV administration, plasma C_max_ concentration of **MG3** was on average 395 ng/mL. After oral administration, C_max_ of **MG3** was on average 213 ng/mL, achieved 6 h post-dosing (Table 7). **MG3** was still detectable in the plasma up to 72–96 h post oral dosing (Figure 3). 

**MG3** showed high clearance and high volume of distribution (Table 7). The half-life of the compound was similar after oral or IV administrations (10.7 and 7.7 h, respectively), indicating that the rate of absorption did not affect the disposition processes of the compound. The absolute bioavailability of **MG3** was complete, i.e., 100%, better than that reported for other 4 aminoquinoline [53], such as isoquine (ISO), des-ethyl isoquine (DEI), N-tert-butyl isoquine (NTBI) (68%) or AQ13 (70%) [58]. The high volume of distribution vs. relatively low plasma concentration, suggestive of high tissue distribution, is in line with the high efficacy observed in vivo and confirms previous PK studies of **MG3** in the mouse [22]. This is indeed a characteristic common to many drugs of the 4-aminoquinoline family and it contributes significantly to their potent antimalarial efficacy [45,53,58,59]. As for bioavailability, the 4-aminoquinoline including **MG3** and lysosomotropic weak bases, a major factor controlling their tissue distribution and pharmacokinetics is their ability to be sequestered in lysosomes [60].

#### 3.3.3. Tissue Distribution, Plasma Protein Binding and Blood Partitioning of MG3

The tissue distribution of **MG3** was then investigated after single oral administration at 120 mg/kg to female Sprague Dawley rats. **MG3** was detected in all the analyzed tissues up to 96 h post dose. At all-time points the concentration in tissues was higher than in plasma. The highest concentration of **MG3** was detected in lungs and ovaries, but in the interval 4–48 h high levels were also measured in the liver, spleen, and kidney (Figure 4). This is quite like what was observed for CQ or hydroxychloroquine [61]. For amodiaquine, the main organs of accumulation after oral administration were kidney, liver, red bone marrow and spleen [62]. 

Moreover, the extent of binding of **MG3** to plasma proteins in human and in the animal species (rat and dog) was moderate, ranging between 67% and 84% (corresponding to percentages of free fractions ranging between 33% and 16%). The binding was higher in human (range between 81.3% and 84.3%) and was > dog > rat. 

This seems to be consistent with what reported for other 4-aminoquinolines [53]. The blood to plasma ratio of **MG3** concentration in human blood up to 30 min was approximately 1, suggesting that **MG3** does not preferentially associate with blood cells. However, preliminary experiments indicate that **MG3** accumulates inside RBC 3–4 times better than CQ. By comparing the PK profile of normal versus *P. berghei* infected CD1 mice, it was found that **MG3** (10 mg/kg) is absorbed after oral treatment with a peak after 120 or 60 min, respectively. Exposure resulted slightly higher in the infected group compared to the control AUClast 61,959 min*ng/mL compared to 41,067 min*ng/mL of the controls; C_max_, however, were about 145 ng/mL in both the groups (0.49 µM), confirming previous studies with CQ [52].

#### 3.3.4. Preclinical Toxicology

Preclinical safety data are crucial for supporting the progression of a new compound into clinical development. Firstly, the potential genotoxicity of **MG3** was tested using the Bacterial Reverse Mutation Assay starting from the dose of 15.63 mg/plate to 1000 mg/plate [63,64]. No significant dose-related and reproducible increases in revertant colony numbers of the *Salmonella typhimurium* tester strains TA98, TA100 and in the *Escherichia coli* strain WP2 uvrA were observed with any dose tested, with or without exogenous metabolic activation with rat liver microsome. **MG3** was therefore judged to be non-mutagenic, and the detailed data are shown in Appendix A.

The potential interaction of **MG3** with the cardiac hERG K+ channel was evaluated via [3H]-astemizole binding assay in HEK293 cells stably transfected with the human ERG (ether-a-go-go related gene) [65]; both **MG3** and CQ showed a moderately low interaction with the hERG K^+^channel with K_i_ nM of 553 ± 45 and 2281 ± 223, respectively. The Ki for astemizole was 3.3 ± 0.7 nM. 

The in vivo toxicity of **MG3** was assessed using three animal species standardly used for toxicological studies in vivo, namely rats, dogs, and non-human primate. In the 7-day oral dose range finding (DRF) experiment at 30, 60, and 120 mg/kg/day in rats, no mortality occurred, and no clinical signs were observed in any treated animal. A dose-related decrease in mean body weight gain occurred between Day 1 and Day 4 in males, with trend to recovery only at the two lower doses of 30 and 60 mg/kg/day and in females at 30 mg/kg/day. Body weight loss was observed in females administered 60 and 120 mg/kg/day. Minimal to moderate decrease in food consumption was recorded in all treated animals. At clinical chemistry on Day 8, slight to moderate increase in ALT and AST were seen in two out of three females treated with 120 mg/kg/day. Slight increase in AST was also observed in a few males treated with 60 and 120 mg/kg/day. At necropsy, all treated animals showed good general condition. The systemic exposure to **MG3** was measured on Days 1 and 7. The females were generally more exposed (at most by a factor of two) than the males, both following a single and repeated treatment (Table 8). Negligible accumulation was observed on Day 7 in animals administered 30 and 60 mg/kg/day, whilst at 120 mg/kg/day the systemic exposure in each gender was about five-fold higher than on Day 1.

Females appeared more sensitive than males to the effects of the compound, an observation that could be explained by differences in the systemic exposure values of **MG3**, which were about two-fold higher in females than in males. Based on the current results, the oral dose of 120 mg/kg/day was considered as the maximum tolerated dose (MTD) in rats, which is a result comparable and even better than other aminoquinoline derivatives [42,52,53]. The response of dogs to the treatment was quite different. The doses of 15 and 30 mg/kg/day were associated with mortality one week after last treatment due to acute renal failure and severe in-life and clinical chemistry findings. These results were in line with previous data from compounds of the same family [45], so it was concluded that the dog are not suitable as non-rodent species to assess the toxicity of **MG3**. The experiments were then performed in NHP by treating one male and one female monkey with incremental doses of 25, 50 and 75 mg/kg/day of **MG3** for three days/dose (Phase I). The dose of 75 mg/kg/day was associated with mortality and severe CNS-related clinical signs including decreased activity, tremors, loss of coordination, and recumbency. Upon these data, subsequently one/sex monkey was administered with **MG3** at 50 mg/kg/day for five consecutive days. This fixed dose was tolerated with limited clinical signs consisting of daily episodes of emesis, a lower food intake and variations in liver enzymes. Based on these observations, the maximum tolerated dose (MTD) in this study is 50 mg/kg/day. At this dose, the AUC_(0–24)_ was 802 and 989 ng.h/mL and the C_max_ 67.5 and 72.7 ng/mL, for the male and the female, respectively, based on Day 5 values (Table 9). After 5-day repeated daily oral administration in NHP, despite the different t_max_ and t1/2 observed in the two monkeys, no gender-related differences in the exposure to **MG3** was observed neither in terms of C_max_ nor daily AUC_(0–24)_.

## 4. Conclusions

In summary, we provide evidence that **MG3** is a promising candidate for further development for the treatment of uncomplicated malaria in combination with other drugs with different mechanisms of action. Here, the initial observation on the in vitro and in vivo activity of **MG3** have been extended to include a complete non-formal preclinical evaluation of the ADME-Tox properties of this molecule.

**MG3** belongs to the 4-aminoquinolines family of compounds, such as CQ, and shares the same mechanism of action, that is the inhibition of haem detoxification by intraerythrocytic parasites [24]. In addition, and different from CQ, **MG3** is active in vitro against a large panel of *P. falciparum* laboratory strains and field isolates known to be resistant to CQ. In preliminary 72 h assays against IPC5202, a *P. falciparum* strain carrying the artemisinin resistance-associated mutation R539T of the Kelch13 gene [7], **MG3** showed an IC_50_ of 10 nM, which is the lowest compared to that of the other strains tested in vitro, indicating good activity against ARM-resistant parasites. However, since resistance to the ACT partner drugs of the same class of **MG3**, such as piperaquine or amodiaquine, is present in the Great Mekong Subregion and may spread to Sub-Sahara Africa, further studies with **MG3** using modified chemosensitivity assays, such as the piperaquine survival assay, and genotyped field isolates [66,67] would be required to predict whether **MG3** could be efficaciously used in combination against drug-resistant parasites. At present, from the isobologram analysis, the activity of **MG3** seems to be additive in vitro to that of artemisinin derivatives, artesunate and DHA used in combination, with a not ignificant trend toward antagonism. This agrees with previous reports from both our group [23] and others [35,47] on different endoperoxides tested in vitro in association with 4-aminoquinolines molecules. These in vitro data taken together with the prolonged half/life of **MG3** and its high bioavailability suggest a possible application of **MG3** in both double and triple ACT. Recent data indicate that triple ACT regimens are as efficacious as ACT and may be employed in areas with high prevalence of artemisinin-resistant parasites [12]. 

This manuscript also reports that the synthesis of **MG3** was optimized from the previous described protocol [24] and is now suitable for a scale-up. The in vivo and in vitro ADME-Tox studies excluded any hepatotoxicity potentially due to the presence of the pyrrolizidine ring moiety. Moreover, **MG3** showed a low/moderate clearance in human, with low or negligible interference with P450 isoforms. Unchanged **MG3** was the major metabolic component in human and rat. No human specific metabolites were seen. **MG3** was not mutagenic and showed a moderately low binding affinity with the cardiac hERG K+ channel. After DRF experiments in rats (7-day oral toxicity study) no mortality and no clinical evidence of liver toxicity up to 60 mg/kg were seen. These data confirm the rational design of **MG3**: the substitution at the bridgehead carbon does not allow the formation of toxic pyrrole containing metabolites. In rats, the oral dose of 120 mg/kg/day was considered as the maximum tolerated dose. Toxicity experiments were also conducted in non-rodent species, dogs, and non-human primate. As expected for this class of molecules, **MG3** was not well tolerated in dogs, whereas in NHP at the end of the 5-day oral toxicity study, the maximum tolerated dose was 50 mg/kg/day. These values are in line and even better than those obtained with CQ or the other quinoline in preclinical development [45,52,53,58]. In rats, the half-life of **MG3** is similar after IV and oral administrations (7.7 and 10.7 h, respectively), with high volume of distribution and 100% absolute bioavailability.

In conclusion, following our initial design idea of introducing in the 4-aminoquinoline structure a bulky, strongly basic, and lipophilic bicyclic moiety (as the quinolizidine or the pyrrolizidine ring), it was possible to overcome the CQ resistance mechanisms by preventing the metabolic oxidative dealkylation, which affects the usual dialkylaminoalkyl chains of many CQ analogues. This approach was successful, leading to the discovery of **MG3,** a novel chemical entity ready to enter the development phase as a potential antimalarial candidate. 

## Data Availability

Not applicable.

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
