# Peer review of "Favorable Preclinical Pharmacological Profile of a Novel Antimalarial Pyrrolizidinylmethyl Derivative of 4-amino-7-chloroquinoline with Potent In Vitro and In Vivo Activities"

_biomolecules, 2023, doi:10.3390/biom13050836_

Round 1

Reviewer 1 Report

This manuscript, "Favourable preclinical pharmacological profile of a novel anti-malarial pyrrolizidinylmethyl derivative of 4-amino-7-chloro-quinoline with potent in vitro and in vivo activities," by Basilico et al. is, in a word, solid. The manuscript is well-written and (in the opinion of this reviewer) worthy of publication in Biomolecules.

There are a few minor writing fixes to be considered by the authors:

line 29:
4-aminoquinolines drugs
should be
4-aminoquinoline drugs

Line 72:
common and unique scaffold
should be
common scaffold

Line 495:
Table S3
should be
Table S4

Line 562:
from MG3 metabolism
should be
from the MG3 metabolism

Line 660:
Table S4
should be
Table S5

Line 710:
The Table Title should contain the species studied

Line 716:
drug with different mechanism of action
should be
drugs with different mechanisms of action

Reviewer 2 Report

In this work, the authors report the optimized and safer synthesis of MG3, and additional in vitro and in vivo characterization. They found that the pharmacological profile of MG3 is in line with those obtained with CQ or the other quinolines in use and seems to possess all the requirements for a developmental candidate. The work is interesting and important for new drug discovery, but there are still many problems to be improved.

All the ml should change to mL.

The structure is vague and needs to be redrawn.

Spaces need to be added between data and units.

Clarifying the mechanism of action can deepen people's understanding of drugs, and the mechanism of action of compound MG3 should be clarified.

The author claims that the route can be used for large-scale production of MG3, but the synthetic amount of these compounds in the article is only gram level, which is not consistent with large-scale production.

The synthesis part lacks the carbon spectra data of these compounds, and the spectra and liquid phase data of these compounds are not provided, so the purity of these compounds cannot be determined. 

Author Response

"Please see the attachment

Reviewer 3 Report

This article addresses an important topic regarding the search of new antimalarial drugs. This study is very rich, comprehensive, clearly introduced, well designed and presented.

About the form, many editing errors remain in the manuscript: missing dot between “P” and “falciparum” (or other species), “in vivo” and “in vitro” sometimes in italics and sometimes not, line 230 missing space between artesunate and (ASN), references are missing line 57-59, line 235 “P. chabaudi chaboudi” instead of “P. chabaudi chabaudi”, line 495 wrong table number S3 instead of S4 (please verify other numbering).

Line 76, about AQ13 development: this molecule is not present in MMV portfolio, could you clarify its status (development by other company, arrested development, or other reasons, …).

Line 206: could you precise the nature of resistance of the strain TM902 (which resistance, which molecular markers, etc…).

Line 206: precise the K13 mutation in IPC5202 (If I am wright, R539T).

About field isolates collected in patients: if the information is available, it will be very informative to provide the K13 genotype for these strains in supplementary data and to separate them in Table 2 into 2 sub-groups (ART-resistant and ART-sensitive).

Regarding toxicology experiments, please precise where or by whom in vivo experiments on dog and primates have been realized, and the approval of these experiments by ethic comities.

Regarding in vitro antiplasmodial evaluation, MG3 have been evaluated on many strains (for laboratory or field) with diverse drug resistance genotype, which is a real bonus for this study. It is only a pity to not explore further MG3 activity in the context of artemisinin resistance (which is expanding in endemic regions). Indeed, artemisinin resistance involves many special mechanisms with consequences for the assessment of chemosensitivity. Classical IC50 evaluation of MG3 against ART-resistant strains is necessary but not sufficient. It has been previously demonstrated that ART-resistant parasites partially lose sensitivity to other antiplasmodial drugs including chloroquine, but that this phenomenon can’t be evidenced by standard chemosensitivity assay due to specific ART-resistance mechanism (Ménard S, et al. Emerg Infect Dis. 2015 Oct;21(10):1733-41. doi: 10.3201/eid2110.150682). Even if such experiments were not included in the present paper, this topic must be addressed at least in the discussion of the results.

For table 7, could you be more precised in the legend and explain the different parameters (AUC0, tlast, CL, etc….). Same comment for table 8 and 9.

In the conclusion you write that the activity of MG3 is additive to that of artemisinin derivatives, whereas Figure S2 shows an antagonist effect between MG3/ASN and MG3/DHA: please clarify this point.

Author Response

"Please see the attachment

Reviewer 4 Report

The presented manuscript represents a significant research in the field of synthesis of new antimalarial drugs. The data obtained in the work are certainly of great interest to readers, however, before accepting the article for publication, I recommend reflecting additional points:

1. Potential drug targets in the parasite should be discussed

2. Title of section 3.1. worth specifying

3. For a synthesized and purified preparation, I would recommend submitting an NMR spectrum

4. The methodology for conducting animal testing should be specified, including a description of the number of animals at each point.

5. It is necessary to add a special section that reflects the statistical methods used in the work

6. It is necessary to add to each table in the description the exact conditions of the experiment, the error values

Author Response

"Please see the attachment

Round 2

Reviewer 4 Report

The authors have considered all my remarks and I believe this manuscript could be accepted in the present form